Transfection types, methods and strategies: a technical review

Chong Zhi Xiong 1
http://orcid.org/0000-0002-4736-0674 Yeap Swee Keong 2
Ho Wan Yong 1 WanYong.Ho@nottingham.edu.my
1 School of Pharmacy, University of Nottingham Malaysia , Semenyih, Selangor , Malaysia
2 China-ASEAN College of Marine Sciences, Xiamen University Malaysia , Sepang, Selangor , Malaysia
Vassetzky Yegor
Electronic publication date: 2021 Apr 21
Publication date: 2021
Volume: 9
Electronic Location ID: e11165
Received 2020 Jul 22; Accepted 2021 Mar 5
Copyright: © 2021 Chong et al.
Copyright year: 2021
Copyright holder: Chong et al.
License: This is an open access article distributed under the terms of the Creative Commons Attribution License, which permits unrestricted use, distribution, reproduction and adaptation in any medium and for any purpose provided that it is properly attributed. For attribution, the original author(s), title, publication source (PeerJ) and either DOI or URL of the article must be cited.
License URL: https://creativecommons.org/licenses/by/4.0/

Keywords: Methods, Nucleic acids, Controls, Efficiency, Chemicals, Transfection

Funding: Ministry of Higher Education, Malaysia (FRGS) FRGS/1/2018/STG05/UNIM/02/1 and FRGS/1/2014/SG05/UNIM/02/1 This work was supported by FRGS grants by the Ministry of Higher Education, Malaysia, grant numbers FRGS/1/2018/STG05/UNIM/02/1 and FRGS/1/2014/SG05/UNIM/02/1. The funders had no role in study design, data collection and analysis, decision to publish, or preparation of the manuscript.

==============================
Transfection is a modern and powerful method used to insert foreign nucleic acids into eukaryotic cells. The ability to modify host cells’ genetic content enables the broad application of this process in studying normal cellular processes, disease molecular mechanism and gene therapeutic effect. In this review, we summarized and compared the findings from various reported literature on the characteristics, strengths, and limitations of various transfection methods, type of transfected nucleic acids, transfection controls and approaches to assess transfection efficiency. With the vast choices of approaches available, we hope that this review will help researchers, especially those new to the field, in their decision making over the transfection protocol or strategy appropriate for their experimental aims.

Introduction

Transfection is a process by which foreign nucleic acids are delivered into a eukaryotic cell to modify the host cell’s genetic makeup (Kim & Eberwine, 2010; Chow et al., 2016). For the past 30 years, transfection has gained increasing popularity due to its wide application for studying cellular processes and molecular mechanisms of diseases (Arnold et al., 2006; Ishida & Selaru, 2012; Chow et al., 2016). Understanding the molecular pathway of disease allows the discovery of specific biomarkers that may be applied to diagnose and prognose diseases (Ye et al., 2017; Roser et al., 2018). Besides, transfection can be employed as one of the strategies in gene therapy to treat incurable, inherited genetic diseases (Yao et al., 2008; Yamano, Dai & Moursi, 2010; Tomizawa et al., 2013). Today, the advancement in life-sciences technology allows different types of nucleic acids to be transfected into mammalian cells, and these include Deoxyribonucleic acids (DNAs), Ribonucleic acids (RNAs) as well as small, non-coding RNAs such as siRNA, shRNA and miRNA (Borawski et al., 2007; Yamano, Dai & Moursi, 2010; Sork et al., 2016; Shi et al., 2018).

Generally, transfection can be classified into two types, namely stable and transient transfection (Kim & Eberwine, 2010; Stepanenko & Heng, 2017). Stable transfection refers to sustaining long-term expression of a transgene by integrating foreign DNA into the host nuclear genome or maintaining an episomal vector in the host nucleus as an extra-chromosomal element (Lufino, Edser & Wade-Martins, 2008). The transgene may then be constitutively expressed even with the replication of cells (Recillas-Targa, 2006; Kim & Eberwine, 2010).

In contrast, transient transfection does not require integrating nucleic acids into the host cell genome (Riedl et al., 2018). Nucleic acids may be transfected in the form of a plasmid (Nejepinska et al., 2012) or as oligonucleotides (Igoucheva, Alexeev & Yoon, 2006). Therefore, transgene expression will eventually be lost as host cells replicate (Recillas-Targa, 2006; Kim & Eberwine, 2010). Transient transfection is usually applied in short-term studies to investigate the effects of knock-in/-down of a particular gene (Kim & Eberwine, 2010). In contrast, stable transfection is useful in long-term genetic and pharmacology studies in which large-scale protein production is needed (Elgundi et al., 2017).

A vector construct that carries the specific nucleic acids to be transfected can be further divided into either viral or plasmid vector. Viruses and plasmids facilitate the expression of a foreign transgene via the presence of a suitable eukaryotic promoter (Colosimo et al., 2000). A viral vector may trigger an immunogenic response in the host cell while a non-viral vector is comparatively less immunogenic (Hardee et al., 2017). A delivery mechanism is needed to facilitate the transfer of targeted nucleic acids or vector construct into the host cell (Kim & Eberwine, 2010). Some of these entail physical methods while others involve the use of a delivery vehicle, which may be lipid-based (Balazs & Godbey, 2011) or non-lipid based (Jin et al., 2014), to help enhance the contact between vector-vehicle complex with the host cell membrane, thereby facilitating the entrance of the complex into cells (Balazs & Godbey, 2011).

Designing and initiating a transfection assay can be challenging, especially with the vast variety of transfection approaches or strategies to choose from Gharaati-Far et al. (2018), Shi et al. (2018) and Tan et al. (2019). Thus, we aimed to provide a systematic comparison between different strategies, targets, controls, reagents and validation involved in transfection to provide an overview for beginners in this field. This review included an overview of different transfection methods (viral vs. non-viral approaches) and the common types of transfected nucleic acids (DNA, RNA and small RNAs). Other important aspects such as the type and importance of controls, choice of methods for assessing transfection efficiency, and potential factors influencing efficiency were also discussed.

Survey methodology

A systematic literature search based on PRISMA guidelines (Moher et al., 2009) was established to identify relevant published studies or protocols that fit into this review’s scope (Fig. 1). Databases that were employed for the literature search included Scopus, Google Scholar, and PubMed. The keywords being used during the search included “transfection”, “co-transfection”, “chemicals”, “reagents”, “DNA”, “siRNA”, “shRNA”, “miRNA”, “plasmid”, “oligonucleotides”, “efficiency”, “safety”, “cytotoxicity”, “controls” and other related key terms. An initial search returned about 5,000 articles, published protocols, or handbooks from various databases that reported the descriptions or comparisons between different transfection methods, types of transfected nucleic acids, transfection control, transfection efficiency assessment methods and transfection reagents. Two independent reviewers performed article screening and the selection to avoid selective bias. About 500 articles were retained after the first screening to remove duplicated sources or articles that appeared in more than one database. Inclusion criteria of the literature selection include English articles published in the past 30 years and articles or sources that reported the description or comparison of the different transfection methods/protocols/strategies/procedures. About 100 written sources that did not fit the inclusion criteria were removed, and these included letters to the editor, conference paper, and articles with inaccessible full-texts. Around 150 relevant and related published literature or handbooks were selected in the final stage to compare different transfection protocols, types of transfected nucleic acids, transfection controls, methods to assess transfection efficiencies, and transfection reagents in terms of their strengths, effectiveness, safety level and limitations.

Figure 1 The workflow of literature selection using PRISMA flow strategy.

Keywords used during the literature search included “transfection”, “co-transfection”, “chemicals”, “reagents”, “DNA”, “siRNA”, “shRNA”, “miRNA”, “plasmid”, “oligonucleotides”, “efficiency”, “safety”, “cytotoxicity” and “controls”. Inclusion criteria of the literature selection comprised of English written articles or sources which were reported in the past 30 years since 1990 and articles which reported the description or comparison between different transfection protocols, types of transfected nucleic acids, transfection controls, methods to assess transfection efficiencies and transfection reagents in terms of their strengths, effectiveness, safety level and/or limitations. About 150 written sources were used in the qualitative analysis of this technical review.

Transfection approaches

Transfection methods can be generally divided into viral, non-viral, or a combination of both (hybrid) (Fig. 2).

Figure 2 Different transfection protocols that can be divided into viral-based, non-viral based or combination of both (hybrid).

Viral transfection

Viral-based transfection, or more specifically known as transduction, involves using a viral vector to carry a specific nucleic acid sequence into a host cell. Retroviruses, such as lentiviruses are often used for stable transfection (Pfeifer & Verma, 2001; Kim & Eberwine, 2010; Fakhiri et al., 2019). In contrast, adenovirus, adeno-associated virus (AAV) and herpes virus are viral vectors which do not guarantee stable transfection (Lee et al., 2017). As compared to non-viral transfection, viral transduction is widely recognized as a highly effective method to transfect difficult-to-transfect cells such as primary cells (Mali, 2013; Wang, Shang & Li, 2015). Generally, retroviruses can only be used to transfect dividing cells while adenoviruses, AAVs and herpes viruses can be used to transfect both dividing and non-dividing cells (Lee et al., 2017). However, viral transduction is associated with higher cytotoxicity and may pose a risk for viral infection (Kim & Eberwine, 2010; Mali, 2013).

A viral vector usually contains a viral envelope that surrounds and protects the virus. Surface proteins may be present on the surface of certain types of viruses, such as adenovirus, to facilitate contact and communication with the host cell (Maginnis, 2018). Viral genetic materials are enclosed in a capsid, which will be unpacked upon entering the host cell. Unlike the genomes of adenoviruses, AAVs and herpes viruses which are maintained episomally (Hardee et al., 2017; Lee et al., 2017), a retroviral genome is integrated into the host genome (Lee et al., 2017). Generally, adenoviruses and herpes viruses carry double-stranded DNAs, AAVs carry single-stranded DNAs, while retroviruses carry RNAs (Lee et al., 2017). Integrase is a type of enzyme produced by viruses such as retroviruses that facilitate the integration of foreign genetic materials into the host genome (Hindmarsh & Leis, 1999; Tzlil et al., 2003). In retroviruses, RNAs will be reverse transcribed into a double-stranded viral DNA before being integrated into the host cell genome for replication and expression (Hindmarsh & Leis, 1999). Even in non-retroviruses that lack integrase, such as the adenoviruses, integrase may be expressed through genetic engineering technique (Mogler & Kamrud, 2015).

Retroviruses that stably express transgene exhibit lower potential in triggering inflammation than adenoviruses and herpes viruses that usually produce transient transgene expression but inflammation in the host cell (Lee et al., 2017). However, using retroviruses for transfection is associated with a high risk of insertional mutagenesis and gene disruption (Hardee et al., 2017). In contrast, viruses that do not induce stable genome integration have a lower but existent risk of triggering mutagenesis (Fernández-frías, Pérez-luz & Díaz-nido, 2020).

Compared to adenoviruses, AAVs exert lower immunogenicity and pathogenicity in human, which makes them a safer virus for gene therapy (Nayerossadat, Maedeh & Ali, 2012). However, the small packaging capacity (<5 kb) of AAVs limits their applications to deliver large-sized therapeutic genes (Lee et al., 2017). On the other hand, adenoviruses have higher packaging capacities than AAVs and were reported to be able to transduce most cell types (Lee et al., 2017). Among the several commonly used viruses for transduction, herpes viruses are found to have the largest packaging capacity (~150 kb) (Lee et al., 2017; Fernández-frías, Pérez-luz & Díaz-nido, 2020). Besides, herpes virus is said to have strong tropism for neuronal cells and thus, herpes virus has great potential to be employed to deliver specific nucleic acid in treating diseases of the nervous system (Lee et al., 2017).

Non-viral transfection

Physical/mechanical transfection

Non-viral based transfection approach can be further classified into a physical/mechanical method and chemical method. Commonly used physical/mechanical transfection method includes electroporation, sonoporation, magnetofection, gene microinjection and laser irradiation (Du et al., 2018; Hamann, Nguyen & Pannier, 2019; Meng et al., 2019). Electroporation is a commonly used physical transfection method that uses electrical voltage to transiently increase the cell membrane permeability to allow the entry of the foreign nucleic acid (Prasanna & Panda, 1997). This method is commonly employed to transfect difficult-to-transfect cells such as primary cells, stem cells and B cell lines (Jordan et al., 2008; Stroh et al., 2010; Liew et al., 2013; Canoy et al., 2020). However, the use of high voltage may cause cell necrosis, apoptosis, and permanent cell damage (Piñero et al., 1997; Kim & Eberwine, 2010; Mali, 2013). Ultrasound-assisted transfection or sonoporation involves the use of microbubbles technique to create holes on the cell membrane to ease the transfer of genetic materials (Meng et al., 2019), while laser irradiation-assisted transfection uses a laser beam to create small holes on the plasma membrane to allow entry of foreign genetic substances (Pylaev et al., 2019). Like electroporation, both sonoporation and laser-assisted transfection also pose risks of damaging the cell membrane and irreversible cell death (Kim & Eberwine, 2010; Mali, 2013). Comparatively, magnet-assisted transfection, or magnetofection that uses magnetic force to aid in the transfer of foreign genetic materials, appears to be less destructive to the host cell despite its low efficiency (Wang, Shang & Li, 2015). Gene microinjection, on the other hand, involves the use of a specific needle to puncture the cell to inject the desired nucleic acids into the nucleus of the host cell (Chow et al., 2016). However, this technique necessitates specially trained personnel or robotic system that can perform the procedure with high precision to prevent cell damage (Kim & Eberwine, 2010; Chow et al., 2016) and is thereby of great value in clinical applications such as gene therapy (Nikol et al., 1999). Compared to physical or mechanical transfection methods, chemical transfection involves using specially designed chemicals or compounds to aid in transferring the foreign nuclei acid into the host cells (Hamann, Nguyen & Pannier, 2019; Tan et al., 2019).

Chemical transfection

Chemical transfection can be categorized into liposomal-based or non-liposomal-based (Kim & Eberwine, 2010). Examples of the commercially available chemical transfection reagents are summarized in Fig. 3. Liposomal-based transfection reagent is a chemical that enables the formation of positively charged lipid aggregates that could merge smoothly with the phospholipid bilayer of the host cell to allow the entry of the foreign genetic materials with minimal resistance (Kim & Eberwine, 2010; Mali, 2013).

Figure 3 Summary of the commonly used chemical transfection reagents.

Transfection reagents can be generally divided into liposomal or high-lipid based and non-liposomal based reagents. Non-liposomal reagents can be mixed protein-lipid reagents, non-liposomal lipids or non-protein non-lipid reagents like dendrimer. The details of the Invitrogen and ThermoFisher Scientific products can be found at: https://www.thermofisher.com/my/en/home.html. Information on the Qiagen related transfection products can be found at: https://www.qiagen.com/us/. Information on the Promega related reagents can be found at: https://worldwide.promega.com/. For Polyplus related transfection reagents, the information can be retrieved at: https://www.polyplus-transfection.com/. For Sigma–Aldrich or Merck or Roche related reagents, the information can be found at: https://www.sigmaaldrich.com/united-states.html. Information on the Thomas Scientific related chemicals can be retrieved at: https://www.thomassci.com/. Details on the Mirus Bio products can be referred at: https://www.mirusbio.com/. For GeneCopoeia products, further information can be found at: https://www.genecopoeia.com/. Information on ClonTech products can be found at: https://www.takarabio.com/. Details on System Biosciences chemicals can be found at: http://www.excellbio.com/. Information on the products by Strategene can be found at: https://www.agilent.com/. Details of the Fermentas International Inc. can be found at: http://fermentas.lookchem.com/.

On the other hand, non-liposomal transfection reagents can be further divided into several classes, including calcium phosphate, dendrimers, polymers, nanoparticles and non-liposomal lipids (Kim & Eberwine, 2010; Mali, 2013; Valetti et al., 2013). Calcium phosphate is one of the cheapest chemicals used in transfection that involves binding the positively charged calcium ions (Ca2+) with the negatively charged nucleic acids to form a precipitate before being taken up by host cell (Guo et al., 2017). However, the success rate of calcium phosphate transfection is comparatively low and requires prior optimization to achieve high transfection efficiency (Guo et al., 2017). Dendrimers are 3-dimensional, highly branched organic macromolecules that could form complexes with nucleic acids claimed to be superior to calcium phosphate as an alternative non-liposomal transfection reagent (Dufès, Uchegbu & Schätzlein, 2005). However, transfection efficiency using dendrimers is still lower than viral vectors and liposomal reagents (Dufès, Uchegbu & Schätzlein, 2005; Borawski et al., 2007). Cationic polymers could also form complexes with the negatively charged nucleic acids, which aid in the uptake of the genetic materials by cells through endocytosis (Kim & Eberwine, 2010; Mali, 2013). Compared to viral vectors and lipofection, cationic polymers produce less cytotoxicity but are also compromised with lower efficiency (Kim & Eberwine, 2010; Mali, 2013). Recently, nanoparticles are emerging as an alternative option in non-liposomal transfection due to their small size, which enhances the entry of nucleic acids into the host cell (Sandhu et al., 2002). Nanoparticles were reported to cause little cytotoxicity to the transfected cells, but more studies may be necessary to carefully assess its efficiency and long-term safety in clinical applications (Al-Dosari & Gao, 2009). Non-liposomal lipid-based nanocarrier is a type of lipid-formulated nanocarrier that allows fast and effective delivery of nucleic acids into eukaryotic cells (Valetti et al., 2013). Compared to liposomal lipid nanocarriers, this nanocarrier was claimed to be safer, but its preparation is tedious and expensive (Valetti et al., 2013; Meisel & Gokel, 2016).

Hybrid of viral and non-viral transfection methods

The combination of viral and non-viral transfection method is known as a hybrid approach, and the use of such a combined method was reported to produce higher transfection efficiency compared to other transfection methods such as using polyplexes alone (Pinnapireddy et al., 2019). However, a hybrid method appears to be more laborious and costly due to the need for the artificial synthesis of the unique viral-like vectors linked with other chemicals such as glucose (Pinnapireddy et al., 2019) and lipids (Keswani, Pozdol & Pack, 2013). Nevertheless, the hybrid-based transfection approach was also proven to be safe and able to produce stable human cell lines that constantly expressed the proteins of interest (Keswani, Pozdol & Pack, 2013; Pinnapireddy et al., 2019).

Type of transfected nucleic acids

Deoxyribonucleic acids

In transfection, DNAs are normally transported into a host cell via a viral or non-viral vector such as plasmid (Balak et al., 2019; Pang et al., 2019). The plasmid’s basic structure includes a promoter, origin of replication, multiple cloning site, gene of interest, and selection marker (Nora et al., 2018). The origin of replication is needed for plasmid replication, while the multiple cloning site contains unique restriction enzyme cut sites for insertion of foreign genes (Nora et al., 2018). The presence of appropriate eukaryotic promoters such as CMV or EF-1α allows the expression of a foreign gene in the host cell (Wang et al., 2017b). Plasmid DNA may be transfected in the form of linear and supercoiled DNA (Mitchenall et al., 2018). Transfection using supercoiled plasmid DNA will generally produce higher efficiency as compared to linear DNA, which is more susceptible to degradation by exonucleases (Von Groll et al., 2006). Linearized DNA, however, is more recombinogenic and can thus be more readily integrated into the host genome to achieve stable transfection (Von Groll et al., 2006; Hardee et al., 2017).

The use of a plasmid vector does not guarantee constitutive transgene expression nor stable foreign DNA integration into the host genome (Delrue et al., 2018). In other words, as cells divide, a foreign gene will not be constitutively expressed and eventually lost if it has not been integrated into the host cell’s genome (Mali, 2013). Therefore, using an appropriate selection marker such as an antibiotic resistance gene or fluorescence protein co-expressed with the transgene is necessary to select and maintain stably transfected cells in culture (Kaufman et al., 2008). Like plasmid vector transfection, the introduction of a selection marker gene into a viral vector is also useful for selecting stably transduced cells (Tomás et al., 2018).

In comparison to viral DNA transfection, plasmid-based DNA transfection is less immunogenic without the risk of viral integration into the host cell genome (Mali, 2013). However, the transfection efficiency and protein production of plasmid-based DNA transfection is comparatively lower (Oh & Kessler, 2018).

RNA and messenger RNA

Similar to DNA transfection, RNA may be introduced into eukaryotic cells via RNA-based viral or non-viral vectors (Mogler & Kamrud, 2015; Oh & Kessler, 2018; Ylosmaki et al., 2019). In comparison to transfection involving DNAs, RNA transfection might produce higher transfection efficiency as the latter does not require transit across the nuclear membrane (Zou et al., 2010). Without the need for genome integration, transcription, and post-transcriptional processing, RNA transfection may also accelerate the desired protein (Oh & Kessler, 2018; Ylosmaki et al., 2019). The use of messenger RNA (mRNA)-based vectors may also prevent complications due to integration into the host genome, thereby allowing specific, desired proteins to be expressed (Mogler & Kamrud, 2015; Oh & Kessler, 2018). However, protein expression is transient following RNA transfection, and RNAs are comparatively less stable than DNAs, making them more prone to degradation when transported intracellularly (Mogler & Kamrud, 2015; Nowakowski et al., 2017; Oh & Kessler, 2018).

Common and special oligonucleotides (small ribonucleic acids)

Small RNAs are RNA molecules of 18–200 base pairs (bp) in length and possess the ability to regulate post-transcriptional gene regulation and RNA modification (Watson, Belli & Di Pietro, 2019). Examples of small RNAs include microRNAs (miRNAs), small interfering RNAs (siRNAs), short hairpin RNAs and piwi-interacting RNA (piRNA) (Watson, Belli & Di Pietro, 2019).

microRNAs and piRNAs are both endogenous and single-stranded small RNAs. miRNAs (18–25 bp) are involved in post-transcriptional regulation of downstream mRNAs by inhibiting the targeted mRNA or interfering with its translation initiation (Wilczynska & Bushell, 2015). On the other hand, piRNAs (24–30 bp) take part in transposon silencing and post-transcriptional regulation (Chuma & Nakano, 2013). Similar to miRNAs and piRNAs, siRNAs also play a role in regulating post-transcriptional gene expression (Allison & Milner, 2014). siRNAs are normally 20–24 bp in length, which may be expressed as endogenous or exogenous double-stranded small RNAs (Allison & Milner, 2014). shRNA is a type of endogenous, double-stranded small RNA with a hairpin loop (Mcintyre et al., 2011). shRNA may bind to the complementary sequence on an mRNA to degrade it (Mcintyre et al., 2011).

One should determine its experimental need before deciding on the appropriate small RNA molecule for transfection-related functional assay. For instance, siRNA is highly specific to only one target, whereas miRNA has the potential to regulate multiple downstream targets (Lam et al., 2015).

Today, various types of short-length oligonucleotides (Table 1) may be artificially synthesized to imitate the small RNA molecules for functional studies of the knock-in/-down/-out effects of these small RNA molecules. The commonly used oligonucleotides can be grouped into either mimic or antagonist (Bell & Micklefield, 2009). A mimic is a small RNA-based oligonucleotide (may be piRNA, miRNA, or siRNA) that has a structure which enables it to bind to a targeted mRNA to inhibit its function, resulting in translational repression of a specific gene (Fu, Jacobs & Zhu, 2014; Ahmadzada, Reid & McKenzie, 2018; Edvard Smith & Zain, 2019). In contrast, an antagonist is an oligonucleotide that will bind to the complementary small RNA strand such as miRNA to antagonize its activity, thereby increasing the targeted gene expression (Edvard Smith & Zain, 2019; Fu, Chen & Huang, 2019).

Table 1 List of commonly used oligonucleotides in the small RNA transient transfection work.

	Common small RNA oligonucleotides	Agomir/antagomir	Locked nuclei acid (LNA) oligonucleotide	
Product examples (Origin)	(1) Qiagen (Valencia, CA)
(2) ThermoFisher Scientific (Ambion) (Waltham, Massachusetts, USA)
(3) IDT (Coralville, IA)
(4) GeneCopoeia (Rockville, USA)
(5) Dharmacon (Cambridge, UK)
(6) GenePharma (Shanghai, China)	(1) GenePharma (Shanghai, China)
(2) RiboBio (GuangZhou, China)
(3) Creative Biogene (NY, USA)	(1) Exiqon (Vedback, Denmark)
(2) IDT (Coralville, IA)
(3) Sigma–Aldrich (Saint Louis, MO, USA)	
General structure		
Number of strands	May be single or double-stranded (duplex)	Agomir: double-stranded
Antagomir: single-stranded	Mimic: triple-stranded (One guide strand and two passenger strands)
Antagonist: single-stranded	
Strengths	(1) Commonly available and cheaper
(2) Easy to be introduced into host cells	(1) Stable structure and effective in action
(2) Higher affinity towards cell membrane,
(3) Longer interfering effect which may last up to 6 weeks	(1) Effective and stable structure
(2) May or may not require a transfection reagent	
Efficiency comparison		Superior than normal mimics/inhibitors	LNA oligonucleotides > common oligonucleotides Mixture of LNA+OMe oligonucleotides > common oligonucleotides	
References	(Cheng et al., 2005; Bell & Micklefield, 2009; Jensen, Anderson & Glass, 2014; Jin et al., 2015)	(Krützfeldt et al., 2005; Chen et al., 2015; Merhautova, Demlova & Slaby, 2016; Hu et al., 2017)	(Tolstrup et al., 2003; Chan, Krichevsky & Kosik, 2005; Fabani & Gait, 2008; RNA Functional Analysis Handbook, 2016; Qiagen miRCURY LNA Mimics & Inhibitors & Target Site Blockers Handbook, 2017; Piao et al., 2018)	

With the advancement in the oligonucleotide biosynthesis industry, different types of modified oligonucleotides were also introduced into the market to increase the efficiency of transfecting small RNA oligonucleotides. One of them is the agomirs and antagomirs that are chemically modified to improve their binding affinities to target and block exonuclease activities (Merhautova, Demlova & Slaby, 2016; Hu et al., 2017). An agomir is an artificially modified double-stranded miRNA mimic designed to exert higher target repression activity than conventional miRNA mimics (Krützfeldt et al., 2005; Fu, Chen & Huang, 2019). On the other hand, antagomir is a specially designed single-stranded miRNA analog that aims to inhibit a specific miRNA (Krützfeldt et al., 2005; Fu, Chen & Huang, 2019). Both agomir and antagomir were claimed to be more stable, effective, specific, and have higher binding affinity to the host cell membrane than the normal mimics or antagonists (Krützfeldt et al., 2005; Chen et al., 2015; Hu et al., 2017).

Locked nucleic acids (LNA) is another type of modified oligonucleotide that possesses an extra methylene bridge in at least one of its nucleotides to enhance the stability of its ribose ring structure (Grünweller & Hartmann, 2007). Its locked ribose structure makes LNA shorter than the commonly used oligonucleotides, thereby enabling it to show higher efficiency, stability, and binding affinity than traditional oligonucleotides (Tolstrup et al., 2003; Chan, Krichevsky & Kosik, 2005; Fabani & Gait, 2008; Qiagen miRCURY LNA Mimics & Inhibitors & Target Site Blockers Handbook, 2017). The application of LNA-based oligonucleotides has been reported in various biochemical or functional assays that involved the deliveries of small RNA molecules such as siRNA (Elmén et al., 2005), miRNA (Roberts et al., 2006) and piRNA (Lee et al., 2011). Some LNA-based transfection requires no transfection reagent (Hillebrand et al., 2019), which could minimize secondary effects from the reagents during transfection.

Combination of different transfected nucleic acids or co-transfection

Co-transfection is a process in which more than one type of nucleic acid is being introduced into the eukaryotic cell (Fig. 4) (Hannig & Jany, 2013; Li et al., 2014; Russell, Stefanovic & Tscharke, 2015). Some examples of the combinations include multiple plasmid DNAs (Karda et al., 2019; Bauler et al., 2020), siRNA and plasmid DNA (Kwak, Han & Ahn, 2019; Setten, Rossi & Han, 2019), and multiple miRNAs into the same cell (Seo et al., 2015; Tsukita et al., 2017).

Figure 4 Applications of co-transfection in biotechnology and life sciences research.

(A) Multiple plasmids co-transfection is useful in generating a hybrid vector and is for protein–protein interaction studies. (B) Multiple small RNAs co-transfection is popular in RNA interference and functional assay study to evaluate the regulatory effects of the small RNA on the expression of the downstream target. (C) Co-transfecting a small RNA molecule and a plasmid DNA that carries a reporter system can be used to assess small RNA transfection efficiency or to determine the regulatory effects of the small RNA on a specific gene.

Generally, multiple plasmid DNA co-transfection aims to introduce more than one type of foreign genes into the host cells. One of its applications is to produce synthetic viral or hybrid vectors that consist of several plasmid DNA components (Karda et al., 2019; Bauler et al., 2020). An example is the generation of a lentivirus from several plasmid vectors such as transfer, envelope, and packaging vectors in HEK293 cell line (Merten, Hebben & Bovolenta, 2016). Besides, co-transfection of multiple plasmid DNAs can also be applied in protein–protein interaction (PPI) studies to investigate the relationships between one protein to another (Deriziotis et al., 2014; Vyncke et al., 2019). PPIs can be assessed using physical measurement based on energy transfer from one donor protein to a recipient protein (Deriziotis et al., 2014) or via chemical measurement in which the interactive activity between an expressed protein with another protein can be detected via an appropriate reporting system upon stimulation (Vyncke et al., 2019). The latter luciferase-based method is known as bioluminescence resonance energy transfer (BRET), and it serves as an alternative method to fluorescence resonance energy transfer in studying PPIs (Khamlichi et al., 2019). Co-transfection of multiple plasmids can also be applied in transfection which involves the delivery of plasmids that encode Cas9 protein and guide RNA to the host cell for genome editing using the CRISPR/Cas9 genome engineering system (Gam et al., 2019). Apart from using multiple plasmids, a bicistronic vector, a vector capable of expressing two different genes joined using an internal ribosome entry site with only one promoter, is another way to deliver different genes to a host cell (Li & Wang, 2012).

Co-transfection of small RNA and plasmid DNA can be used to assess transfection efficiency (Horibe et al., 2014). This is feasible by introducing a plasmid containing a luciferase reporter gene in which its 3′ UTR can be recognized by a specific small RNA such as miRNA, which then allows binding by the small RNA to inhibit luciferase expression (Repele & Manu, 2019). On the other hand, co-transfection of small RNA and plasmid DNA is also essential in RNA interference (RNAi) study to determine the regulatory effects of a specific small RNA such as siRNA on the specific gene expression in a host cell (Keller et al., 2018; Ervin et al., 2019). Small, non-coding RNAs such as siRNAs are well known for their epigenetic regulation capabilities or, more specifically, regulation of specific gene expression post-transcriptionally (Zhao & Zhang, 2018). siRNA mimics and a plasmid DNA carrying the specific gene linked to a luciferase reporting system can be simultaneously co-transfected into a target cell (Salim, Islam & Desaulniers, 2020). Successful knockdown of specific gene expression by the siRNA mimic will lead to a measurable decrease in luciferase activity (Peralta-Zaragoza et al., 2016; Shyamasundar, Lim & Bay, 2016).

Co-transfection may also involve different small molecules such as miRNAs, which is useful in studying the effects of small RNAs on the targeted host cells (Tsukita et al., 2017). For instance, if the introduction of a miRNA mimic can affect the expression of a specific downstream gene in a particular cell type, it is then expected that simultaneous transfection of its inhibitor sequence into the same cell would reduce the post-transcriptional regulatory effect exerted by the miRNA mimics sequence alone (Seo et al., 2015).

As compared to transfection of a single nucleic acid type, co-transfection, which involves the transfer of multiple nucleic acids into the same cell, is generally more challenging as not all nucleic acids types can be effectively transferred, and this may depend on the transfection method and cell type involved (Kretzmann et al., 2018).

Transfection controls

The use of control in a transfection experiment is vital for determining the effect and efficiency of transfection reagents and nucleic acids used (Godbey, Zhang & Chang, 2008; Yang, Qiu & Xu, 2011; Jin et al., 2015; Ayub, Ling & Cun, 2016).

Generally, both plasmid transfection and oligonucleotide transfection experiments require a positive control, a negative control, an un-transfected control, and a mock transfection control (Yang, Qiu & Xu, 2011; Jin et al., 2015; Ayub, Ling & Cun, 2016; Stepanenko & Heng, 2017; Gharaati-Far et al., 2018). A positive control is a DNA or RNA that had previously been proven to cause known effects to the transfection experiment, such as affecting the expression of a specific downstream genetic target (Ayub, Ling & Cun, 2016). A positive control is needed during the initial stage of a transfection work to establish an optimized transfection protocol, and afterward, the positive control can act as a reference to be compared to the experimental group. On the other hand, a negative control is used to confirm if an intended gene expression change in the host cell is attributable to the transfection instead of other causes. In plasmid DNA transfection, a negative control can be a reaction that lacks the DNA, the transfection delivery vehicle, or absence of both with only the host cell (Liang, Knight & Jolly, 2013). In small RNA transfection, a negative control contains a non-homology sequence, which is usually a scrambled sequence that shares the same nucleotide length and composition as the target sequence but not homologous to any known mammalian gene (Yang, Qiu & Xu, 2011; Ayub, Ling & Cun, 2016).

An un-transfected control involves the culture of cells without transfection reagents and nucleic acids, which act as a control for basal information about the host cell, including viability, phenotype, and more importantly, the baseline expression level of target gene without the impact from transfection (Godbey, Zhang & Chang, 2008; Gharaati-Far et al., 2018). Mock transfection refers to transfection without the genetic target or nucleic acids, which allows the assessment of effects resulting from transfection reagents such as background auto-fluorescence noise (Hunt et al., 2010; Gharaati-Far et al., 2018). In plasmid transfection experiments, the use of an empty plasmid control, which contains only the vector backbone with the exclusion of the transgene, is recommended as the mock transfection control (Jin et al., 2015; Kamens, 2015).

Assessing transfection efficiency

Assessing the efficiency of transfection is vital, especially in functional studies which require high transfection efficiencies to warrant post-transcriptional regulation of specific downstream targets (Weilin et al., 2004; Marjanovič et al., 2014; Peng et al., 2017). A variety of strategies may be chosen to assess transfection efficiency, where each of these is associated with different pros and cons (Table 2).

Table 2 Comparison of the different methods for the assessment of transfection efficiency.

	Fluorescence microscopy	Real time PCR (qPCR)	Plasmid reporting system	Flow cytometry	Others, eg: western blot & Immunofluorescent staining	
Descriptions	Use of fluorescently tagged molecules to confirm transfection has taken place	Directly measures the expression level of nuclei acids post-transfection	Indirectly measures transfection efficiency via a luminescence measurement or β-galactosidase level	Quantify the number of fluorescently labelled transfected cells	Indirectly monitor the transfection efficiency via detection or quantification of downstream targeted protein expression	
Quantitative or qualitative findings	Qualitative or semi- and quantitative	Quantitative	Quantitative using luminescence reporting system	Quantitative	Semi-quantitative (western blot); Qualitative/ Semi-quantitative (Fluorescent microscopy after immunostaining); Quantitative (Flow cytometry after immunostaining)	
Advantages	Easy and fast	Allows quantification of the transfection efficiency	Allows quantification of the transfection efficiency	Allows quantification of the transfection efficiency	Allows simultaneous assessment of the regulation of downstream protein targets	
Disadvantages	Inability to
distinguish signals originating from interior vs exterior of cells	Expensive and laborious especially in transient transfection where regular monitoring of the transfection efficiency is needed	Reporting system within plasmids not offered by all manufacturers.	Expensive, and laborious	Expensive, laborious and time-taking; false negative results may be obtained due to inappropriate sampling time; non-specific protein binding	
References	(Marjanovič et al., 2014; Mastropietro et al., 2015; Peng et al., 2017)	(Godbey, Zhang & Chang, 2008; Jiwaji et al., 2010; Thomson et al., 2013; Werling et al., 2015)	(Nasim & Trembath, 2005; Jiwaji et al. 2010; Horibe et al., 2014; Marjanovič et al., 2014; Yun & DasGupta, 2014)	(Ho et al., 2006; Marjanovič et al., 2014; Homann et al., 2017)	(Weilin et al., 2004; Buchwalow et al., 2011; Granieri et al., 2012; Liang, Mason & Lam, 2013; Bass et al., 2017; Peng et al., 2017; Shi et al., 2018)	
Note:

Different approaches for assessing transfection efficiency.

Real-time polymerase chain reaction (qPCR) is a quantitative approach for assessing transfection efficiency via direct measurement of the expression level of specific foreign nucleic acids in the cell or other intracellular nucleic acids that could be affected by exogenous nucleic acids such as miRNAs (Godbey, Zhang & Chang, 2008; Jiwaji et al., 2010; Thomson et al., 2013). In the case of transient transfection, qPCR should be performed after each transfection to ensure good transfection efficiency before proceeding to downstream experiments (Thomson et al., 2013).

Co-transfection with plasmid reporter system is another strategy that can be used to assess transfection efficiency by expressing specific reporter proteins such as luciferase or β-galactosidase (Nasim & Trembath, 2005; Jiwaji et al., 2010; Yamano, Dai & Moursi, 2010; Horibe et al., 2014). Taking luciferase reporter system in small RNA interference (RNAi) study as an example, successful transfection of miRNA is indicated by downregulation of luciferase activity, which is due to mRNA degradation as a result of the binding of miRNA to the 3′-end of the transcribed luciferase mRNA (Aldred, Collins & Trinklein, 2011).

Fluorescence microscopy is another common, easy and rapid method to assess transfection efficiency (Marjanovič et al., 2014; Peng et al., 2017). It usually involves using a vector that carries a fluorescence reporting gene or oligonucleotides tagged with fluorophores to allow fluorescence detection (Faltin, Zengerle & Vonstetten, 2013). However, fluorescence microscopy provides only qualitative or semi-quantitative measurement of transfected efficiency, which can be determined using specialized software such as ImageJ (Jensen, 2013). In contrast, flow cytometry allows for more precise quantitation of the cells that express a specific fluorescent gene to assess transfection efficiency (Ho et al., 2006; Marjanovič et al., 2014; Homann et al., 2017).

Another way of assessing transfection efficiency is via monitoring specific protein expression post-transfection (Alabdullah et al., 2019; Mori et al., 2020). Introduction of a transgene into the cell may alter the expression of a protein encoded by the transgene or other genes in the cell (Kim & Eberwine, 2010). Likewise, transfection of small RNAs may also regulate the expression of specific downstream genetic targets in the host cell (Liang, Hart & Crooke, 2013). Immunoblotting and immunofluorescent staining may be employed to assess changes in the expression of protein post-transfection. The use of specific antibodies for binding to targeted proteins are vital in both methods, where the latter requires the use of secondary fluorescently labelled antibodies that bind to the primary antibodies to detect the protein of interest (Sograte-Idrissi et al., 2020). On the other hand, in immunoblotting, horseradish peroxidase (HRP)-conjugated secondary antibodies can be used to bind to the primary antibodies for specific protein detection (Lin et al., 2016). Immunoblotting allows semi-quantification of protein expression while immunofluorescence staining allows detection via fluorescent microscopy or quantification via flow cytometry. The assessment of transfection efficiency via examination of specific protein expression is more reproducible and straightforward (Zeitelhofer et al., 2007; Homann et al., 2017). However, the issue of non-specific proteins binding inherent from the use of antibodies (Liang, Mason & Lam, 2013; Niikura & Kitagawa, 2016) and the likelihood of obtaining false-negative results, which may be caused by untimely assaying of protein expression (Brunner et al., 2000) remain as the drawbacks of using these methods.

Factors influencing transfection efficiency

The efficiency of chemical transfection depends greatly on a few factors such as type of reagents used, the origin and nature of target cells, and an optimum DNA to reagent ratio chosen (Table S1) (Gharaati-Far et al., 2018; Shi et al., 2018; Wang et al., 2018). In this section, we will review past research that reported the influence of these factors on various transfection strategies’ efficiency.

Factors influencing the efficiency of viral transfection

Viral vectors such as lentiviruses are useful in gene therapy due to their ability to carry large-sized nucleic acid and deliver their targets to both non-dividing and dividing cells (Karda et al., 2019). A few factors had been shown to potentially affect efficiencies of viral transduction, such as target cell types, type of promoter used, the concentration of vector, and condition of transduction medium used (Table 3) (Ikeda et al., 2002; Denning et al., 2013).

Table 3 Factors influencing the efficiencies of viral and physical/mechanical transfection methods.

	Virus transduction	Physical/mechanical transfection methods	
Electroporation	Laser beam	Nucleic acid injection	Ultrasound-assisted	Magnet-assisted	
Factors affecting transfection efficiencies	(1) Virus type/generation
(2) Cell type
(3) Promoter present in viral vector
(4) Presence of other substances during transduction	(1) Electroporation condition (duration and voltage used)
(2) Number of electric pulses
(3) Cell type
(4) Electroporation buffer composition
(5) Size and concentration of nucleic acids	(1) Laser condition (power density and duration)
(2) Number of laser pulses
(3) Cell type	(1) Number of injection repeat
(2) Amount of injected nucleic acids
(3) Size, shape and coating of needle
(4) Cell type	(1) Ultrasound exposure condition
(2) Number of pulses
(3) Cell culture condition
(4) Cell type
(5) Amount of nucleic acids used	(1) Magnetic condition (oscillating or static conditions)
(2) Modification of the magnetic nanoparticles
(3) Number of pulses
4) Cell type	
References	(Haas et al., 2000; Ikeda et al., 2002; Denning et al., 2013)	(Potter & Heller, 2003; Molnar et al., 2004; Yao et al., 2009; Hornstein et al., 2016; Chopra et al., 2020; Sherba et al., 2020)	(Stevenson et al., 2006; Terakawa et al., 2006; Tsen et al., 2009)	(Wells et al., 1998; Dahlhoff et al., 2012; Mellott, Forrest & Detamore, 2013; Chow et al., 2016)	(Ogawa, Tachibana & Kondo, 2006; Kinoshita & Hynynen, 2007; Zhou et al., 2012)	(Mykhaylyk et al., 2007; Fouriki et al., 2010; Kardos & Rabussay, 2012; Wang et al., 2014)	

In an in vitro study that evaluated the use of human immunodeficiency virus (HIV) and equine infectious anemia virus (EIAV) for gene transduction, varying degrees of efficiencies were observed between the two viruses on different cell lines from human, hamster, cat, dog, horse and pig. The transduction efficiency using HIV was generally better than EIAV on most cell types, while the infectivity of both viruses on rodent cells was weak (Ikeda et al., 2002). In the same study, the type of promoter was also suggested as a factor that may influence transduction efficiency whereby HIV containing the internal promoter EF-1α in replacement of CMV promoter exhibited higher transduction efficiency on murine and rat cells (Ikeda et al., 2002). Besides, viral concentration was suggested as another factor influencing efficiency. Among a few other parameters tested in the evaluation by Haas et al. (2000), namely HIV lentiviral vector construct containing different accessory proteins, presence/absence of fibronectin fragment and addition of polycations protamine sulfate into transduction medium on human cord blood and embryonic kidney cells, only viral titer appeared to be directly associated with viral transduction efficiency.

On the other hand, the condition of the transfection medium may also affect transduction efficiency. For instance, the use of fetal bovine serum was shown to yield better transduction efficiency than bovine calf serum during transduction (Denning et al., 2013). Likewise, polycations such as DEAE-dextran were shown to minimize repulsion forces between negatively-charged cells and facilitated viral transduction (Denning et al., 2013).

Factors influencing the efficiency of chemical transfection

Type of chemical transfection reagents

The choice of a suitable transfection reagent can depend on several factors, including the type of transfected nucleic acids and complexity of the transfection (single or co-transfection) (Attractene Transfection Reagent Handbook, 2008; HiPerFect Transfection Reagent Handbook, 2010; Kim & Eberwine, 2010; Trans IT-X2® Dynamic Delivery System Protocol, 2019). Some reagents such as Effectene and TransIT-X2 are specially dedicated for plasmid DNA transfection (Homann et al., 2017; Ormeño et al., 2020), while some reagents such as Lipofectamine RNAiMAX are more suited for transfection of small oligonucleotides (Jensen, Anderson & Glass, 2014; Wang et al., 2018). Several reagents had also been suggested to be suitable for co-transfection of multiple DNA plasmids, whereas a distinct group of reagents is catered for mixed DNA/small RNA molecules co-transfection (Table S1) (Duportet et al., 2014; Kretzmann et al., 2018; Shi et al., 2018; Wang et al., 2018; Tan et al., 2019).

Type of chemical transfection reagents

Primary and stem cells

Primary mammalian cells are generally less susceptible to transfection than other cell types due to its finite lifespan and limited expansion capacity (Oh et al., 2007; Yamamoto et al., 2017; Alabdullah et al., 2019). Non-liposomal-based reagents were shown to be superior than liposomal reagents in transfecting primary human cells, including PEC (Young et al., 2002), HASMC and HAEC (Kiefer et al., 2004), primary human myoblast (Arnold et al., 2006) and AGS (Gharaati-Far et al., 2018). In contrast, liposomal-based reagents such as Lipofectamine and DharmaFECT families were reported to produce higher transfection efficiencies than non-liposomal reagents in transfecting other primary human cells such as the primary umbilical cord vein endothelial cells (HUVEC) (Hunt et al., 2010) and BM-MSC (Cheung et al., 2018).

In a study that involved transfection of plasmid DNA into HUVEC, the use of non-liposomal reagents including Effectene and FuGENE 6 produced better transfection efficiency (34% and 33%) than the liposomal reagent DOTAP (18%) (Young et al., 2002). In another study that transfected HUVECs with plasmid DNA (Hunt et al., 2010), liposomal-based reagents, however, demonstrated higher transfection efficiencies (~38% at 48 h for Lipofectamine LTX and ~23% at 48 h for Lipofectamine 2000) as compared to non-liposomal reagents such as Effectene and FuGENE 6 or HD (all <20% at 48 h). Lipofectamine LTX and Lipofectamine 2000 were reported to show similar cytotoxic effects to HUVEC but the cytotoxic effects of other reagents were not reported. The efficiencies of liposomal-based vs. non-liposomal-based reagents in transfecting HUVECs could not be concluded in these studies, mainly due to differences in the range of reagents used. However, consistent observations that transfection efficiencies remained below 40% regardless of the reagents used had undoubtedly implicated primary cells as a hard-to-transfect cell type.

Human primary stem cells is another well recognized hard-to-transfect cell type, where poor efficiency and low cell viability remain as the greatest challenge in transfecting this cell type (Ervin et al., 2019; Tan et al., 2019). In 2015, Wang, Shang & Li (2015) reported that transfection reagents such as Lipofectamine 2000 and XtremeGENE HP produced very poor transfection efficiencies (<6%) in human periodontal ligament stem cell as compared to the positive control lentiviral vector that achieved ~95% of transfection efficiency. Compared to the magnetic assisted transfection technique employed in the same study, the latter showed greater transfection efficiency (~11%) with lower toxicity. In another study that involved human bone marrow mesenchymal stem cell (hBM-MSC), Lipofectamine LTX was shown to produce the best transfection efficiency (at least three times higher) than other reagents such as TransIT-2020, Lipofectamine 3000 and polyethylenimine (PEI) but presented low cell viability (<50%) (Cheung et al., 2018). Comparatively, a better outcome may be attained by using TransIT-2020 reagent that was shown to achieve around 30% efficiency, with recovery up to 90% of the cells and attainment of about 95% of cell stemness (Cheung et al., 2018). Another example of hard-to-transfect stem cell is induced pluripotent stem cells (iPSCs). In a study which compared different non-viral methods for transfecting human iPSC-derived cardiomyocytes (hiPSC-CMSs), Lipofectamine STEM was shown to produce superior transfection efficiency (up to 32%) as compared to other reagents (Lipofectamine 3000, Lipofectamine 2000 and the non-liposomal PEI-based reagents Transporter™ 5 and PEI25) that produced efficiencies below 20% (Tan et al., 2019).

As a general guideline, using cells from the early passage was recommended to achieve good transfection efficiency, especially for transfection that involves primary or stem cells (Young et al., 2002; Covello et al., 2014; Wang, Shang & Li, 2015). Another interesting observation was that 37 °C was the optimal incubation temperature that could help reach higher transfection efficiency in primary cells (Young et al., 2002). This phenomenon could be because 37 °C is the optimal culture temperature for mammalian cells (Wang et al., 2017a). Meanwhile, chemical transfection appeared to be less appealing than viral and physical transfection for transfecting primary cells, especially in human primary stem cells.

Human vs animal cells

The origin of cell lines, such as human vs. animal cell lines, may also contribute to varying degrees of efficiencies when transfected using the same transfection reagents under similar conditions (Kiefer et al., 2004; Kim & Eberwine, 2010; Maurisse et al., 2010). In a study which involved transfection into smooth muscle cells from human and rat (Kiefer et al., 2004), most of the transfection reagents were reported to show higher efficiencies in transfecting rats smooth muscle cells (A-10 SMCs) as compared to human aorta smooth muscle cells (HASMCs). Among the seven transfection reagents tested (DAC-30, DC-30, Lipofectin, LipofectAMINEPLUS, Effectene, FuGene 6 and Superfect), FuGENE 6 was concluded to produce the best transfection efficiency in both HASMCs and A-10 SMCs, but the efficiency was 4 fold lower in the former (~10% vs. ~40%). In both cell lines, SuperFect produced the highest cytotoxic effects, followed by DAC-30 and Lipofectamine Plus, while FuGENE 6 was considered comparatively safe to the cell lines (Kiefer et al., 2004).

In another study which compared the transfection outcome between different cell lines from human and animal origin, pig tracheal epithelial cells (PTE) was shown to be more efficiently transfected than the human tracheal epithelial cell line (HTE) by transfection reagents including Effectene, Lipofectamine Plus and PEI (Maurisse et al., 2010). Transfected HTE also exhibited lower viability than PTE post chemical transfection (Maurisse et al., 2010). Combined findings from the two cited studies (Kiefer et al., 2004; Maurisse et al., 2010) suggested that probably animal cell lines may be more efficiently transfected than human cell lines.

Cell lines derived from the same species

The performance of different transfection reagents could also vary in different cell lines derived from the same species. For instance, liposomal reagents were reported to exhibit higher efficiencies than non-liposomal reagents in transfecting various immortalized human cell lines, including HEK293, MDA-MB-231, MCF-7, A549, A673, HCT116, HeLa, HepG2, JU77, Huh-7 and HL-60 (Borawski et al., 2007; Ooi et al., 2016; Sork et al., 2016; Shi et al., 2018; Wang et al., 2018). The same trend was also observed in transfection into animal cell lines involving P16 and PTE (Maurisse et al., 2010), PC12 (Covello et al., 2014), bMDAM (Jensen, Anderson & Glass, 2014), E14 and R1 cells (Tamm et al., 2016).

However, non-liposomal reagents were shown to produce higher transfection efficiencies than liposomal-based reagents in cell lines, including MCF-7, HepG2, 4T1, HCT116 and HEK293 (Yamano, Dai & Moursi, 2010; Homann et al., 2017). In animal cell lines, non-liposomal reagents were only reported to produce higher efficiency in a study that involved transfection into Z3 cell line (Sandbichler, Aschberger & Pelster, 2012).

In general, lipid-based or liposomal reagents demonstrated higher transfection efficiencies in most studies that involved immortalized human and animal cell lines. In some cell lines such as MCF-7, HepG2 and HEK293, the comparisons between liposomal-based vs. non-liposomal-based reagents were not conclusive, which may suggest that some cell lines are less selective to the type of reagents for effective transfection, thereby offering researchers a wider choice of approaches to opt for transfecting these cells.

On the other hand, non-liposomal reagents appeared to be comparatively safer than the liposomal-based reagents (Sandbichler, Aschberger & Pelster, 2012; Cheung et al., 2018). In both primary and immortalized human cell lines, liposomal reagents were reported to produce more significant cytotoxic effects than non-liposomal reagents (Kiefer et al., 2004; Yamano, Dai & Moursi, 2010; Homann et al., 2017; Cheung et al., 2018; Wang et al., 2018). Consistent with the observations, liposomal-based reagents were also reported to cause higher toxicities in the animal cells, the studies by Sandbichler, Aschberger & Pelster (2012) and Covello et al. (2014).

Adherent vs. suspension cells

Suspension cells are commonly known as being more challenging to be transfected than adherent cells due to reduced potential attachment of transfection complex to the suspension cells’ surface (Basiouni, Fuhrmann & Schumann, 2012). However, a study comparing the efficiencies of Xfect, Lipofectamine 2000, Nanofectamin, TransIT-X2 and TransIT-2020 showed that all reagents except Xfect showed higher efficiency transfecting suspension cells as compared to adherent cells (Tamm et al., 2016). However, the reasons underlying the opposing observations remain largely unclear and may be further explored in the future.

The ratio of nucleic acids to transfection reagents

The ratio of nucleic acids to transfection reagents also plays a role in influencing transfection efficiency (Arnold et al., 2006; Shi et al., 2018). In a study involving primary human myoblasts, the effect of transfection efficiencies was compared using different nucleic acid ratios to transfection reagents including FuGENE 6, Effectene, and ExGen 500 (a PEI-based reagent) (Arnold et al., 2006). One remarkable finding from the study was that transfection efficiency might not necessarily correlate directly to the reagent volume used. For instance, the ratio of 2 µg of DNA to 5 µL of FuGENE 6 reagent was shown to produce the best transfection efficiency, whereas lower or higher DNA to reagent ratios did not improve the efficiency (Arnold et al., 2006). A similar finding was also observed in another study involving transfection into human gastric adenocarcinoma cell line whereby the optimum transfection efficiency was not achieved using the highest transfection reagent to DNA ratio volume tested among a range of combinations (Gharaati-Far et al., 2018).

The use of disproportionate and high transfection reagent volume would cause unwanted cytotoxicity that reduces the overall transfection outcome (Arnold et al., 2006; Gharaati-Far et al., 2018). Therefore, determining an appropriate nucleic acid to reagent ratio is an important step in initiating a new transfection study to achieve high transfection efficiency and low cytotoxicity (Gharaati-Far et al., 2018; Shi et al., 2018).

Other factors

Serum-reduced or serum-free media are normally recommended in transfection involving cationic transfection reagents such as Lipofectamine (Wallenstein et al., 2010), HiperFect (Diener et al., 2015) and EndofectinMax (Shi et al., 2020). Such transfection activities entail cationic liposome-DNA complexes formation, which requires the interaction between positively-charged liposomal molecules and negatively charged nuclei acids (Son, Tkach & Patel, 2000). As such, the presence of negatively charged molecules in serum could potentially affect the complex interactions, thereby affecting transfection efficiency (Simoes et al., 2000; Misra et al., 2013). However, the presence of 10% serum was found to result in higher transfection efficiencies in MCF-7, HeLa, C2C12 and MC3T3 transfected with FuGENE HD, jetPEI, Lipofectamine 2000 and Arrest-In (Yamano, Dai & Moursi, 2010). It was suggested that a minimal amount of serum in transfection could improve inter-surface interaction between the transfection complexes and its host cell surface by modulating the zeta potential of the transfection complexes (Yamano, Dai & Moursi, 2010).

On the other hand, freeze-thawing of transfection reagents was suggested as another potential factor that may influence transfection efficiency. Lipofectamine 2000 reagent that underwent at least one freeze thaw cycle produced superior transfection efficiency in the cell lines HEK293, Neuro2a, C2C12 myoblasts and myotubes, hTERT MSC, SMA and HepG2 as compared to its nonfrozen control, without a compromise in cell viability (Sork et al., 2016).

A possible explanation for this is that freeze-thawing could enhance molecular rearrangement dispersion, thereby allowing a higher dispersion rate that allows maximal contact between the nucleic acids to form more transfection complex (Sork et al., 2016). However, more studies are needed to support this practice as freeze-thaw process was also suggested to cause recrystallization that could damage some chemicals’ structures (Cao et al., 2003).

Factors influencing efficiency of physical/mechanical transfection

Factors affecting transfection efficiencies of the physical or mechanical transfection depends largely on the principles underlying these methods (Table 3). For instance, the electroporation technique relies on an electrical field to increase the host cell membrane’s permeability to internalize foreign nucleic acids (Sherba et al., 2020). As such, the voltage and duration during the electroporation process are important factors that determine the success of electroporation. Prolonged electroporation with high voltage applied could potentially lead to cell damage and reduce transfection efficiency (Molnar et al., 2004). The electro-transfection efficiency can also be improved by increasing the number of electric pulses, but this may reduce the cell viability (Chopra et al., 2020). On the other side, the electro-transfection efficiency is dependent on the type of cell used and the electroporation condition should be optimized whenever a new cell type is going to be electro-transfected (Potter & Heller, 2003). Some cell like T lymphocyte, might be poorly transfected even a standard electroporation condition is being applied while electro-transfection of fibroblasts could produce generally good transfection outcome (Potter & Heller, 2003).

The electroporation buffer’s composition is another critical parameter that influences transfection efficiency. ATPase inhibitor such as lidocaine in the electroporation buffer was reported to improve cell viability post-electroporation while the use of K+-based buffer resulted in better transfection efficiency than Mg2+-based buffer. Mg2+ ions were hypothesized to play a key role in activating ATPases to restore the ionic homeostasis post-electroporation to minimize cell death but could potentially reduce the transfection efficiency (Sherba et al., 2020). Therefore, a suitable recipe of electroporation buffer consisting of various components should be optimized to ensure a balance between transfection efficiency and cell viability post-electroporation is well-maintained (Sherba et al., 2020). Besides, the size and concentration of the vector should be carefully selected during electroporation. Large plasmid size was reported to reduce electroporation transfection (Molnar et al., 2004; Hornstein et al., 2016). In another study, the efficiencies of electroporation transfection in dental follicle cells gradually improved as plasmid concentrations increased in the range between 0.02 and 0.26 μg/μl. However, not only did efficiency not improve, cell survival also declined post-electroporation with further increment of plasmid concentration to 0.3 μg/μl (Yao et al., 2009). In circumstances when low electric pulse number is used, a high plasmid DNA concentration can be applied to compensate the low number of electric pulses to achieve a good electro-transfection efficiency (Chopra et al., 2020).

In laser-assisted transfection, density and duration of laser-irradiation used to deliver foreign nucleic acids into the host cells would primarily affect the success of transfection (Stevenson et al., 2006; Tsen et al., 2009). Higher laser power and prolonged exposure to laser do not necessarily warrant good efficiency. In fact, moderate (Stevenson et al., 2006) to low laser power and short exposure to laser-irradiation (Tsen et al., 2009) showed superior efficiency and sustained cell survival compared to more prolonged exposure and higher laser power during laser-assisted transfections. Besides, the efficiency of laser-assisted transfection can be enhanced by increasing the number of laser pulse (Terakawa et al., 2006). The use of different cell types during laser-assisted transfection would also show different transfection efficiencies. For instance, in a study (Terakawa et al., 2006) which compared the laser-assisted transfection efficiencies between five different cell types, it was demonstrated that laser-assisted transfection of HeLa cells produced highest transfection efficiency while CHO and human glioma cells showed relatively lower laser-assisted transfection efficiencies.

Ultrasound-assisted transfection involves creating tiny pores on the host cell membrane to facilitate the delivery of nucleic acids, including DNAs and RNAs (Tomizawa et al., 2013). Similar to the former strategies, exposure duration, pulse number, and density of ultrasound irradiation were also reported to be proportionately associated with transfection efficiency up to a threshold value (Ogawa, Tachibana & Kondo, 2006; Kinoshita & Hynynen, 2007; Zhou et al., 2012). Beyond a tolerance limit, cell survival and transfection efficiency may decline. Likewise, increasing the amount of nucleic acids was also shown to improve transfection efficiency of ultrasound-assisted transfection (Zhou et al., 2012). In the same study, transfecting human 293T cells cultured in suspension was claimed to be easier than transfecting the same cell type grown as adherent culture (Zhou et al., 2012). However, this observation did not agree with the findings from another study, which reported no significant difference in the number of the transfected rat cells cultured in either suspension or monolayer conditions (Kinoshita & Hynynen, 2007). Remarkably, the latter study also reported that rat cells grown as monolayer culture maintained higher cell viability post-transfection (Kinoshita & Hynynen, 2007). Nevertheless, discordant findings from these two studies suggest that the optimal culture condition for sonoporation may vary with different cell lines used. In another study (Ogawa, Tachibana & Kondo, 2006), it was shown that sono-transfection efficiency varied with different cell types being used and the sono-transfection of Hela and T-24 cell lines showed better efficiencies than PC-3, U937 and Meth A cell lines. Therefore, the choice of cell type is another factor which could influence sonoporation transfection efficiency (Ogawa, Tachibana & Kondo, 2006).

Magnetic-assisted transfection or magnetofection is another non-viral physical transfection approach and it can be used to deliver either metal-coated nucleic acids complex or magnetic-conjugated AAV vector into the host cell (Mykhaylyk et al., 2007). There are two ways by which magnet can assist in cell transfection, namely, oscillating and static condition. In oscillating magnetofection, a magnet is placed near the host cell and moved side-by-side to drag metal-coated nucleic acids nanoparticles complex onto the host cell surface to facilitate endocytosis mechanically. In contrast, static magnetofection does not involve such steps (Fouriki et al., 2010). The oscillating system was claimed to demonstrate better transfection efficiency than the static system and sometimes superior to chemical transfection (Fouriki et al., 2010). Besides, the distance between cells and magnet could also affect transfection efficiency, whereby a shorter distance will generally result in higher transfection efficiency (Fouriki et al., 2010). Interestingly, surface modification of magnetic nanoparticles had also been shown to improve the efficiency of magnet-assisted transfection. For instance, modifying the surface of Fe3O4 nanoparticles for gene delivery with PEI enhanced plasmid DNA binding affinity to host cell, thereby improving the transfection outcome (Wang et al., 2014). Other factors which could influence the efficiency of magnetofection include cell types being used (Mykhaylyk et al., 2007) and number of pulses (Kardos & Rabussay, 2012). For example, the use of magetofection to transfect human lung epithelial cells might produce better transfection efficiency than Jurkat cells (Mykhaylyk et al., 2007), and the use of double magnetic pulse could increase the transfection efficiency by twice as compared to single magnetic pulse in delivery the specific plasmid DNA in vivo (Kardos & Rabussay, 2012).

Gene injection involves direct delivery of a desired nucleic acid material into the host cells’ nucleus via injection. This method serves as a good alternative when cell transfection is challenging, especially when genetic modification of the host cell is required (Chow et al., 2016). Like other non-viral genes delivery methods, there is no single method that can fit all different cell types (Mellott, Forrest & Detamore, 2013) and the selection of a suitable cell type and nucleic acid size are vital to ensure the success of gene injection. For example, a previous study has reported the successful generation of transgene mice cell line that expressed the cre recombinase (size ~1,000 bp) (Dahlhoff et al., 2012). On the other hand, in a in vivo study, the number of mice muscle fibers expressing transgene correlated with the number of injections and the dosages of the administered plasmid (Wells et al., 1998). This was further supported by another in vitro study, which reported that the number of the host cells expressing a reporter gene was closely correlated to the amount of nucleic acids injected into the cells (Chow et al., 2016). Besides, the size, shape and the presence of additional coating on the microneedles would also affect the efficiency of gene injection as it has been reported that small size microneedles (<10 μm) coated with microparticles were able to suffessfully deliver the desired cargo to the stratum corneum layer of the skin (Mellott, Forrest & Detamore, 2013).

Conclusion

There is no single or universal transfection strategy that is apt for all cell lines and experimental aims. Apart from the experimental budget and the availability of required facilities, the choice for an optimal transfection approach or strategy depends on factors including the type and origin of cells and the form of nucleic acids to be transfected. It is also essential to consider factors that may affect transfection efficiency and cytotoxicity to the host cells and how these parameters may be assessed accurately and conveniently. Likewise, the inclusion of appropriate controls in a transfection experiment is equally important to allow a fair and unbiased assessment of the experimental outcome.

Supplemental Information

Supplemental Information 1 Comparison of the transfection efficiencies of commercially available transfection reagents and transfection conditions reported in different studies (n = 20).

Cell lines details: AGS: Human gastric adenocarcinoma cell line; A549: Human adenocarcinomic alveolar basal epithelial cell; A673: Human ewing sarcoma cell line; A-10 SMCs: Rats smooth muscle cells; bMDM: Bovine monocyte-derived macrophages; C2C12: Mouse myoblasts; C3H10T1/2: Mouse stem cells; HAECs: Human aorta endothelial cells; HASMCs: Human aorta smooth muscle cells; hBM-MSC: Human bone marrow mesenchymal stem cells; HCT116: Human colorectal carcinoma cell line; HEK: Primary human epidermal keratinocytes; HEK293: Human embryonic kidney cells 293; HeLa: Human cervical cancer cell; Hep G2: Human hepatocellular carcinoma cells; hiPSC-CMS: human induced pluripotent stem cells derived cardiomyocytes; HL-60: Human leukemia cell line; hPDLSCs: Human periodontal-ligament stem cells; hTERT MSCs: Human telomerase reverse transcriptase mesenchymal stem cells; Huh-7: Human liver cell line; HUVECs: Human umbilical cord vein cells; JU77: Human primary myoblast and human lung mesothelioma cells; MCF-7: Oestrogen responsive breast cancer cell line; MC3T3-E1: Mouse preosteoblasts; MDA-MB-231: Triple negative breast cancer cell line; MESCs: Sickle cell disease transgenic mice embryonic stem cell; Neuro2a: mice neuroblastoma cells; Panc 04.03: Human pancreas epithelial cancer cell line; PECs: Primary endothelial cells; PTE and THE: Primary pig and human tracheal epithelial cells; PT-30: Human epithelial precancer cells ; P16: Primary embryonic pig fibroblasts; REF: Embryonic rabbit ear fibroblasts; SH-SY5Y: Human neuroblastoma cells; SKOV3: Human ovarian cancer cell line; SMA: Spinal muscular atrophy fibroblast cells; UMR: Rat osteosarcoma cell line; U87MG: human glioblastoma cells ; Z3: Embryonic zebrafish cell line; 4T1: Mouse mammary carcinoma; 16HBE14o- and CFBE41o-: Immortalized human bronchial epithelial cells. Abbreviations for transfection related chemicals/methods: AT: Attractene; DAC: 3β-[N-(N,N’-dimethylaminoethane)-carbamoyl]; DC: 3β-[(N,N’-dimethylaminoethane)-carbamoyl]; DFT1: DharmaFECT1; DFT2: DharmaFECT2; DFT3: DharmaFECT3; DFT4: DharmaFECT4; DOTAP: DTP; EF: ExpressFect; EFT: Effectene; EG500: ExGen 500; E5: Escort IV; FGHD: FuGENE HD; FG6: FuGENE 6; GJ: GeneJammer; HPT: HiperFect; IN: INTERFERin; JP: JetPrime; JPI: JetPEI; LF: Lipofectin; LP+ve: Lipofectamine plus; LP2000: Lipofectamine 2000; LP3000: Lipofectamine 3000; LTX: Lipofectamine LTX; Matra-A: Magnetic-assisted transfection; NaF: Nanofectamine; NFN: Nanofectin; NF: Nucleofection; OF: Oligofectamine; PEI: Polyethylenimine; RNAiMAX: Lipofectamine RNAiMAX; SFT: SuperFect; TF: TurboFect; TKO: TransIT-TKO; TLT1: TransIT-LT1; VF: ViaFect; XG: X-tremeGENE; XG-HD: X-tremeGENE-HD; XG-9: X-tremeGENE-9; X2: TransIT-X2; 293: TransIT-293; 2020: TransIT-2020; “=”: Similar; “>”: More superior.

Click here for additional data file.

Additional Information and Declarations

Competing Interests

Author Contributions

Data Availability

The authors declare that they have no competing interests.

Zhi Xiong Chong conceived and designed the experiments, performed the experiments, analyzed the data, prepared figures and/or tables, authored or reviewed drafts of the paper, and approved the final draft.

Swee Keong Yeap conceived and designed the experiments, performed the experiments, analyzed the data, prepared figures and/or tables, authored or reviewed drafts of the paper, and approved the final draft.

Wan Yong Ho conceived and designed the experiments, analyzed the data, prepared figures and/or tables, authored or reviewed drafts of the paper, and approved the final draft.

The following information was supplied regarding data availability:

No raw data was generated from this literature review.

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
