# Peer review of "Transfection types, methods and strategies: a technical review"

_PeerJ, doi:10.7717/peerj.11165_

## Round 0.1 · original submission · Major Revisions

· Academic Editor

Major Revisions

Please follow the reviewers' remarks to improve the manuscript.

Reviewer 1 ·

Basic reporting

The manuscript by Chong et al provides an overview of the transfection protocols used in modern laboratory practice. Transfection is a routine method utilised in a wide variety of cross-disciplinary studies, thus this topic is of broad interest and within the scope of the journal. Most of the literature available on this topic is focused on specific products or methods, while an unbiased summary of the field is needed. To my knowledge, such a summary has not been done recently, thus the review by Chong et al is relevant and up to date. The review cites recent publications representing a wide range of opinions in the field. The references are selected based on a stringent and unbiased criteria, the sources are adequately cited. The introduction clearly defines the purpose of the article and its target audience, the conclusions are logical. The review is definitely thorough and comprehensive, the wealth of information is carefully summarized, the text is overall logical and coherent, though some points might be improved. I would recommend this article for publication after a revision.

The following steps could be taken to strengthen the article:

Major English language revision. Grammar mistakes are frequent and the word choice is often imprecise, making the message hard to understand. Some examples include lines 47,60,79,88,325,587.

Paying attention to basic concepts and terminology. According to the introduction, the review is aimed at the wide variety of readers, including those new to the field of transfection. To fulfill this aim the review should accurately introduce the terminology and explain the underlying principles of the methods described before making any technical comparisons. Unfortunately, this is now missing. The article could be useful for the readers who already have some experience in the field, but is largely unsuitable for those unfamiliar with transfection principles.

The part “Common and special oligonucleotides (small ribonucleic acids, RNA)” is largely unclear. It would be useful to briefly introduce the readers to the structure of the small RNAs described (e.g. whether they are single-stranded or double stranded) and the molecular mechanisms of their activity in cells before making the comparison of their efficiency. “Common mimics/inhibitors” should be defined more clearly, possibly with some examples.

Minor remarks:

Figure 1 is not very informative and can be either omitted or expanded (e.g. some examples of each transfection type could be added);

The content of the tables sometimes duplicates the text in the main body of the article (e.g. Table 1 and line 194). Please, paraphrase;

In general, the tables are now very hard to read. Making the text in the tables more concise (e.g. by using bullet points instead of the plain text) would be much appreciated;

The descriptions of chemical structures (e.g. the part about LNAs, agomir/antagomir, etc.) are hard to understand and may require illustrations;

Experimental design

no comment

Validity of the findings

no comment

·

Basic reporting

1. In the review by Zhong et al. the authors describe different transfection methods. The topic of the review is indeed of a great interest to scientific community as transfection is a widely used method with a variety of protocols and applications. This article indeed has the potential to be useful for a novice scientist who is starting to explore field of transfection. However, the organization of the entire manuscript can be improved as in the current state the review is a bit superficial and inaccurate. The main concern is that authors do not explain well the different processes and mechanisms that are foreseen in a technology of transfection. Transfection includes delivery, expression and possible integration, and these processes are not necessarily dependent on each other. The notion of integration is not explained well and sometimes is used inappropriately, e.g. in many cases authors refer to plasmid DNA transfection as a method for stable transfection with integration, which is not always the case.
2. The English language needs proof-reading to ensure clear understanding of the review. Some examples where the language could be improved include lines 39, 52, 61, 70, 72, 79, 100, 112, 200, 275, 349, 371, 373, 455 – the current phrasing is incorrect or makes comprehension difficult. Here are some examples:
Line 52 integration in, not between
Line 100 Commonly used physical/mechanical transfection methods includes
3. A brief introduction into the chapters of the review is missing. I would suggest to introduce to the readers that “first we are going to describe X, then Y followed by Z …”

Experimental design

1. Please describe in more details survey methodology. Were PRISMA guidelines followed? If so, please consider slightly modifying your Figure 2 intro classical PRISMA flow diagram (http://prisma-statement.org/PRISMAStatement/FlowDiagram), so your survey can be easily followed and repeated. Please indicate considered years of publications. What were the inclusion criteria that are briefly mentioned on line 79?
2. Figure 1 is described by one phrase “transfection can stable or transient” and in this way is superfluous. Please consider either providing a more detailed classification to visualize “science of transfection” and to help understanding the field or removing the figure. Figure 3 can be improved to make it more informative. Methods can be sorted accordingly to their prevalence. For example, ballistic bombardment is probably not the first method to use among physical methods and it is not even mentioned in the text. Figure 4 can be structured better to make it easier to understand.
3. Lines 46-54. This paragraph is a little bit confusing. Please either explain the notion of vehicle and vector before, or describe the difference between stable/transient transfection in terms of delivery in general with or without integration into the genome. I would also suggest providing examples when transient or stable transfection is applied, i.e. for what purposes.
- Lines 46-48 In this phrasing the message is confusing. Does the “use of a vehicle or vector to carry the targeted nucleic acids into the nucleus” will result “in the stable integration of the foreign nucleic acids”? The delivery and integration are separate processes not necessarily dependent on each other. Please correct.
- Lines 51-52 In this phrasing the message is incorrect. What do authors mean by “vehicle to transport the foreign nucleic acids”? How the nucleic acids are transported then? Liposomes, or calcium phosphate can also be named as delivery vehicles and as such does not assume stable transfection.
4. Lines 132-141. The paragraph is again confusing. Do authors talk about viral or plasmid vectors, as they mentioned both and this results in misunderstanding. No attempt to structure the information is seen. Moreover, reading the text it seems that any DNA that gets into the nucleus is integrated (e.g. line 149) as the processes of delivery/expression/integration are not divided by authors. Plasmid vectors do not usually integrate into the host genome. Moreover, integration does not “mean” that vector has a specific promotor. Eukaryotic promotors ensure expression of transgene in host cell, independently of integration. Finally, eukaryotic resistance gene and FP marker is not a common feature for all vectors. Please consider explaining in more details the structure and principles of viral and plasmid vectors, highlighting their main characteristics systematically, and explaining the processes of delivery, expression and possible integration separately.
5. Lines 167-169. Small RNAs is a big group of different functional RNAs. Please detail about their nature, principles of work and what they are used for.
6. Lines 211. Talking about combination of plasmid DNAs, authors do not mention the option of bicistronic vectors.
7. Line 277. It is not clear what do authors meant by un-transfected control? A simple cell culture without any changes? How it would help assessing the transfection? Authors say that it gives information “whether the absence of transfection-related reagents or materials would exert any effects to the transfection outcomes”. How the absence of transfection itself and/or transfection reagent will have an effect on transfection outcome? Please reformulate.
8. Lines 309-319 Fluoresce analysis of transfection efficiency is not limited to fluorescent dyes linked to DNA, the most common way to assess transfection + expression efficiency is the use of the plasmids that encode for FP.

Validity of the findings

1. The main focus of the review seems to be chemical methods of transfection, other topics related to transfection are much less explored. No protocols/principles of stable transfection were described in the review, which makes it less informative. Authors only briefly mention “viral vectors” without discussing their use, comparing to other methods or highlighting the differences between different viral vectors. Why do authors decide to compare only chemical transfection in terms of efficiency for different cell lines? It would be very useful to discuss also the factors that affect the transfection efficiency of other transfection protocols.

Additional comments

Minor remarks:
Line 40 CRISPR is not the method itself, please correct.
Line 90 Viral transfection is not necessarily stable, e.g. adenoviruses. Please correct.
Line 105 Both necrotic and apoptotic death is possible after electrotransfection (PMID: 14646546).
Line 154 That is true only for DNA-based vectors with a potential to integrate.
Line 161 Again, not all DNA-based vectors are integrated into the genome.
Line 170 Mimic or antagonist of miRNA?
Lines 224-228 Is FRET-based measurement meant there by authors? Investigating PPI has many more approaches besides luciferase reporting system. I suggest either to make it more general, or to name at least some other approaches.
Line 248 Inhibitor or antagonist?
Line 260 plasmid-based transfection is not equal to stable transfection.

---

## Round 0.2 · Minor Revisions

· Academic Editor

Minor Revisions

Thank you for improving the MS. Please address the remaining reviewer comments.

Reviewer 1 ·

Basic reporting

Since the previous revision, the review by Chong at el has been significantly improved. The review is now easier to read, most of the previous remarks have been taken into account. Certain important clarifications have been added. Nevertheless, the article still has some space for improvement.

As a major remark, I would advise the authors to double-check all the essential terms used in the review and make sure all of them are clearly and unambiguously introduced and explained (especially before being referred to elsewhere in the text). For example, in lines 54-62 there is still a “mixture” of the terms “vector” and “delivery vehicle”, which can be highly confusing for the people new to the field.

The minor remarks include:

Lines 36: CRISPR- Cas9 technology is not classified as a transfection strategy.
Line 56: Grammatically Incorrect or incomplete sentence. Ranging from … (to?).
Lines 54-62: the terms “vector” and and “delivery vehicle”.
Line 102: retroviruses include lentiviruses, so “and” can not be used here.
Line 16: other drawbacks and difficulties of viral transfection could be added; the fact that not all viruses have integrases could also be mentioned.
Line 183: please, describe the viral-like vectors in more detail; the technology and its advantages are unclear now; in line 186 cell line examples could be added.
Line 203: a word missing.
Line 299: the gene for guigiing RNA is missing.
Lines 319-320: why in “particular cell types?” (Line 319); Please, specify the cell type in the example which follows.
Line 463: Human primary stem cellS.
Line 511: the cell lines should probably be called “immortalized”, not “secondary”.
Line 551: ...involving primary human myoblasts cells…, “cells” can be omitted.
Line 570: “Reduced or serum-free media are normally recommended…” What is meant by “reduced media” here? The paragraph elaborates only on the serum issue. Please, substitute for “serum-reduced” (if it is the case) or explain otherwise.
Line 590: complexes.
Line 655: correlated with.
Table 2: In the Electroporation column, Cell type should also be mentioned

Experimental design

Study design is adequate and raises no questions.

Validity of the findings

The goals set out in the introduction are close to be fully achieved.

·

Basic reporting

The review by Chong et al. has improved significantly from the previous version. The idea of the review is clear, the material is quite well organized and easy to read. Suffiecient background is provided. Probably, for a person unfamiliar with the field of transfection, this review will serve as a good help to dive into this topic. The Introduction adequately introduce the subject. The review will be ready for publication after some details have been corrected.
Minor corrections:
L. 13. I would use the term “deliver” instead of “insert”
L. 36-37. CRISPR is not a transfection strategy,
L. 54-62. Authors again do not explain the terms of “vector construct” and “transfection vehicle”. It seems that they confuse again the right terms. How DNA plasmid can be a vehicle itself? What if DNA plasmid is delivered via chemical vehicles? Please provide a clear explanation for the terms.
L. 59-62. The comparison makes no sense, one part is about promoter, another is about cellular entry. Viruses also facilitate the entrance into the cell, DNA plasmids within lipid complex can also have eukaryotic promotors.
L. 101-102. Lentivirus IS a retrovirus.
L. 109. Please, specify, which viruses carry an integrase.
P.1.1 Please, specify, what are the drawbacks of using viruses for stable transfection via genome integration. Talking about pros and cons of different commonly used viruses for transduction would be very helpful. Please mention the AAVs, one of the most common tools in gene therapy.
L. 126. Primary cells. If authors want to name difficult-to-transfect cell LINES that are easly electrotransfected, they can name B cell lines (PMID: 32999452).
L. 203. DNA integration into the genome.
L. 299. CRISPR/Cas9 and guide RNA genes.
L. 341. It seems that the described negative control is specific for small RNA transfection experiments, not for DNA plasmid transfection, for example. Please, specify or explain that as well as the notion of non-homology sequence.
L. 342. “interference to the host cells post-transfection” – please reformulate.
L. 357. Not all transfections are performed for the purpose of “post-transcriptional regulation”, please reformulate.
L. 379. in the case of fluorescence reporting gene (one of the most common case), flow cytometry quantifies cells that express a fluorescent gene, not “specially tagged cells”.
L.381-387. Apart immunoblotting, immunofluorescent staining is also commonly used to assess protein expression after transfection.
L. 461. Primary cells, not primary cell line.
L. 463-476. What about iPSCs transfection efficiencies?
L. 478, 481. Again, primary or stem cells.
L. 511, 525. I would suggest to avoid the term secondary and refer to them as “human cell lines”
Table 2. Electroporation: another influencing factor of electroporation condition is the number of pulses and cell type as well as for other transfection methods.
Fig. 2. MagneTofection; please, mention AAV.

Experimental design

The Survey Methodology is consistent with a comprehensive, unbiased coverage of the subject. The review is organized logically into coherent paragraphs/subsections.

Validity of the findings

The goals set out in the introduction are close to be fully achieved.

---

## Round 0.3 · Minor Revisions

· Academic Editor

Minor Revisions

Please revise the paper following the reviewers' suggestions.

Reviewer 1 ·

Basic reporting

The article has significantly improved after the second round of revision. All the reviewers' comments have been taken into account, the article is now much more accurate and new relevant information has been added, as well.
In the revised text, two very minor corrections should be made:
Line 185: Please, explain how the packaging capacity and viral tropism are related (if they are) or remove 'thus' (if they are not).
Lines 132 and 191 completely repeat each other. One should be removed.
Apart from these, the article is ready to be published in its current form and I believe that no further peer revision is required.

Experimental design

no comment

Validity of the findings

no comment

·

Basic reporting

The review is ready to be published, however, I would suggest authors to re-read carefully the manuscript as still some mistakes are remaining.

Line 135: the connection with previous sentence via “thus” is unclear; how packaging capacity and cell tropism are interconnected?
Lines 136-138: repetition, please delete.
Line 322: guide RNA is not equivalent to CRISPR, please correct.
Line 417: immunofluorescent staining does not require the specific antibodies to be fluorescently labelled, the use of secondary fluorescently labelled antibodies is possible – as secondary HRP-conjugated antibodies in immunoblotting. Please, correct.
Line 723: Please use “there is no single method” instead of “there is no a single method”
Figure 2: magneTofection, not magnefofection, please correct.

Experimental design

No comment

Validity of the findings

No comment

---

## Round 0.4 · accepted · Accept

· Academic Editor

Accept

Your paper can now be published in its present form.